

# Description of a common stauromedusa on the Pacific Coast of the United States and Canada, *Haliclystus sanjuanensis* new species (Cnidaria: Staurozoa)

Claudia E. Mills[1], Hannah Westlake[2], Yayoi M. Hirano[3] and Lucília S. Miranda[4]

[1] Friday Harbor Laboratories and the Department of Biology, University of Washington, Friday Harbor, Washington, United States
[2] Department of Biology, University of Victoria, Victoria, British Columbia, Canada
[3] Coastal Branch of Natural History Museum and Institute, Chiba, Katsuura, Chiba, Japan
[4] Department of Zoology, Universidade Federal de Minas Gerais, Belo Horizonte, Minas Gerais, Brazil

Corresponding author
Lucília S. Miranda,
mirandals@ufmg.br

## ABSTRACT

*Haliclystus* "*sanjuanensis*" *nomen nudum* is the most common staurozoan on the west coast of the United States and Canada. This species was described in the M.S. Thesis by Gellermann (1926) and although that name has been in use nearly continuously since that time, no published description exists. Furthermore, the most popular operative name for this species has varied between several related species names over time, resulting in confusion. Herein, we provide a detailed description and synonymy of *Haliclystus sanjuanensis* n. sp., whose distribution is verified from Unalaska Island in the Aleutians (53.4° N, 166.8° W) in the northwest, to Santa Barbara County, California, just north of Point Conception (34.5° N, 120.5° W), in the south. *Haliclystus sanjuanensis* n. sp. is compared with the twelve other described species of *Haliclystus* and illustrations of both macroscopic and microscopic anatomy are provided. *Haliclystus sanjuanensis* n. sp. is unique among species of *Haliclystus* in the arrangement of the bright-white nematocyst spots in its calyx and the pattern of dark stripes running the length of the stalk and up the outside of the calyx.

## INTRODUCTION

Staurozoa is a cnidarian class with about 50 valid species (*Miranda et al., 2018*). They are a group of benthic marine animals generally attached to algae or rocks in intertidal pools or in the shallow subtidal (*Miranda et al., 2018*), generally rare or overlooked but occasionally found in great numbers (*Miranda, Morandini & Marques, 2012*). *Haliclystus James-Clark, 1863* is currently the most diverse genus of the class, with 12 valid species and one additional species assigned as a *nomen nudum* (*i.e.*, a Latin term referring to a species name that was not described in conformance with the *International Commission on Zoological Nomenclature, 1999*), *Haliclystus* "*sanjuanensis*" (see *Miranda et al., 2018*).

This little animal has been known by naturalists and scientists since at least the 1920s and has appeared in most guidebooks to the NE Pacific coastal fauna for the last nearly one hundred years. The species synonymy that we present here is both impressively long and indicative of the persistent confusion over what to call this stalked jellyfish.

*Haliclystus "sanjuanensis"* was first described in the Master's Thesis by *Gellermann (1926)*, a student at the University of Washington Friday Harbor Laboratories in Friday Harbor, Washington, USA. That species name was subsequently used by other visitors to the Friday Harbor Laboratories including *Guberlet (1936, 1949)* and *Hyman (1940a, 1940b)* as if it had been formally described, insinuating itself into the literature. In fact, *Hyman (1940b)* has sometimes been indicated as the author/descriptor of *H. "sanjuanensis"* (*e.g.*, *Kramp, 1961*; *Hirano, 1997*; *Collins & Daly, 2005*), although Hyman was apparently unaware of the necessity to formally describe *H. "sanjuanensis"* and her work did not address the requirements for species descriptions (*e.g.*, a complete description and the designation of a type, see also *International Commission on Zoological Nomenclature, 1999*, Article 13).

Based on morphology, *Gwilliam (1956)* synonymized *H. "sanjuanensis"* (which he had studied in False Bay on San Juan Island, Washington, USA) with *Haliclystus auricula James-Clark, 1863* in his Ph.D. dissertation. Gwilliam never published his taxonomic conclusions, but he used the same identification in an article derived from that dissertation on the neuromuscular physiology (*Gwilliam, 1960*) of *H. "sanjuanensis"* (as *H. auricula*) collected on San Juan Island, Washington. *Kramp (1961)* followed Gwilliam's conclusions and included *H. "sanjuanensis"*, mentioned by *Hyman (1940b)*, in the synonymy of *H. auricula*, and that species name was also selected for the much-used keys for identification of Pacific Northwest invertebrates by *Kozloff (1974)*.

Meanwhile, many authors misidentified specimens of *H. "sanjuanensis"* collected between British Columbia (Canada) and California (USA) as a similar stauromedusa, *Haliclystus stejnegeri Kishinouye, 1899* (*e.g.*, *Ricketts, Calvin & Hedgpeth, 1968*; *Otto, 1976, 1978*; and a number of guidebooks written for the Pacific coast of mainland USA from the 1920s into the 1960s, see synonymies below), perhaps based on the record of this latter species by *Bigelow (1920)* at Port Clarence, Alaska (USA, near the Bering Strait). Contributing to this chaotic scenario, *H. "sanjuanensis"* has also been called *Haliclystus octoradiatus James-Clark, 1863* (*e.g.*, *Eckelbarger & Larson, 1993*). In addition, there is at least one other not-formally-described species of *Haliclystus* in the same region (partially sympatric), also proposed by *Gellermann (1926)* in her Master's Thesis (this species later referred to as *Haliclystus salpinx James-Clark, 1863* in many publications including *Otto, 1976, 1978*; *Mills, 1987*; *Strathmann, 1987*; *Wrobel & Mills, 2003*; *Mills & Larson, 2007*; but this synonymy is yet to be tested and is to be dealt with in a future publication).

On the other hand, there are studies supporting the validity of *H. "sanjuanensis"*. *Hirano (1997)* morphologically revalidated *H. "sanjuanensis"* as different from *H. auricula* based on the presence of prominent white spots (sacs) composed of masses of nematocysts in *H. "sanjuanensis"*; these nematocyst-filled structures are absent in *H. auricula*. *Hirano (1997)* also distinguished *H. "sanjuanensis"* from *H. stejnegeri* and *H. octoradiatus* based on the distribution of the white nematocyst spots in the subumbrella. Phylogenetic

analyses based on ribosomal and mitochondrial genes corroborate that *H. "sanjuanensis"* is clearly distinct from *H. stejnegeri* and *H. octoradiatus* and closely related to the clade composed of *Haliclystus antarcticus* Pfeffer, 1889 and *H. auricula* (*Miranda, Collins & Marques, 2010*; *Miranda et al., 2016a*; *Holst & Laakmann, 2019*; *Holst, Heins & Laakmann, 2019*). The genetic difference (in % based on p-distances) of *H. "sanjuanensis"* to *H. auricula* and *H. antarcticus* is about 3.49–4.31 for 16S and 10.08–11.27 to COI molecular markers (*Holst, Heins & Laakmann, 2019*).

Based on this evidence, and since *"nomina nuda"* and other unavailable names can be made available if they are published again in a way that meets the criteria of availability (*International Commission on Zoological Nomenclature, 1999*; see https://www.iczn.org/outreach/faqs/ "What is a *nomen nudum*"; and Article 13), our aim in this study is to formally describe Gellerman's *Haliclystus "sanjuanensis"* using the species name that she originally applied. We also provide a table comparing *Haliclystus sanjuanensis* n. sp. with the other twelve described species in the genus *Haliclystus*.

## MATERIAL AND METHODS

### Animal collection

We hand-collected about two hundred live specimens of *Haliclystus sanjuanensis* n. sp. at low tides on the south end and west side of San Juan Island and the south end of Lopez Island, San Juan County, Washington State, USA and along the southern shore of Victoria, British Columbia, Canada (approximately 30 km to the west of San Juan Island) for study over a period of many years. Our primary collection locations were the type locality at Cattle Point on San Juan Island, Washington (48.452131° N, 122.961750° W) from the late 1970s–2019 and on the shoreline of the Chinese Cemetery in Victoria, British Columbia (48.406356° N, 123.323467° W) between May and September of 2013–2015. The waters of San Juan County and Cypress Island (the San Juan Archipelago), Washington, are a designated Marine Biological Preserve, with the University of Washington Friday Harbor Laboratories as the managing agency (Washington State House Bill 68, 1923; R.C. W.28.77.230,1969; Revision R.C.W.28B.20.320, 2023); since 1923, the Director of the Friday Harbor Laboratories grants approval of scientific collection of marine plants and animals within the Marine Biological Preserve and C.E.M. has been granted such permission for collection of the marine animals reported in this study. Staurozoa were collected in Victoria, British Columbia, Canada, under Scientific License XR 30 2013–15 issued by the Department of Fisheries and Oceans Canada to Professor Louise Page, academic advisor of H.W. during those years.

A type series of *H. sanjuanensis* n. sp. specimens including holotype and paratypes was collected on July 3, July 15, and July 29, 2019 on low tides at Cattle Point, San Juan Island. Animals were hand-collected from surfgrass or algae around the −50 cm (−1.5 foot) level in shallow water. The animals were transported back to the laboratory and preserved in formalin or ethanol. Formalin-preserved animals were first relaxed in 50% seawater: 50% isotonic magnesium chloride for 30–60 min, and then formalin was gradually added to a final concentration of 10% formalin. Animals preserved in ethanol for future molecular genetic studies were left in clean cold seawater for 6 h; each animal was then lifted out

individually with forceps, blotted briefly on a tissue, and dropped directly into 95% ethanol.

The electronic version of this article in Portable Document Format (PDF) will represent a published work according to the International Commission on Zoological Nomenclature (ICZN), and hence the new names contained in the electronic version are effectively published under that Code from the electronic edition alone. This published work and the nomenclatural acts it contains have been registered in ZooBank, the online registration system for the ICZN. The ZooBank LSIDs (Life Science Identifiers) can be resolved and the associated information viewed through any standard web browser by appending the LSID to the prefix http://zoobank.org/. The LSID for this publication is: urn:lsid:zoobank.org:pub:BF343A8D-DB31-4150-840C-6CDA5A5B121E. The online version of this work is archived and available from the following digital repositories: PeerJ, PubMed Central SCIE and CLOCKSS.

## Morphology and cnidome

Many animals were studied live in small bowls under a dissecting microscope. In general, they did not survive well in the laboratory, remaining in good condition for only a few days, even when fed daily with *Artemia* nauplii. Photos of live animals were taken with an Olympus SZX9 dissecting microscope with Olympus DP72 camera attachment and DP2-BSW software (Version 2.2, Build 6212) and under a Nikon stereomicroscope using a Q Imaging digital camera and Q Capture software.

Observations and measurements of nematocysts from secondary tentacles, white nematocyst spots, gastric filaments, and exumbrellar surface were made on tissues preserved in 10% formalin in seawater solution, squashed in a drop of fresh water on a microscope slide, covered by a cover slip, and gently compressed to further dissociate the cells (*Gwilliam, 1956*; *Mejía-Sánchez & Marques, 2013*). Undischarged capsules of each type of nematocyst were isolated and photographed using brightfield (Zeiss Axioskop, Zeiss Primo Star with Zeiss AxioCam ERc5s) and Nomarski differential interference contrast (DIC) (Leitz Aristoplan) microscopy. Measurements were made using ImageJ software (Version 1.48). Nematocyst terminology follows *Östman (2000)*.

## Histology

Animals for histological analyses were preserved in 10% formalin solution in seawater. We followed the resin technique, described in *Miranda et al. (2017)*. The samples were dehydrated and embedded in glycol methacrylate following the manufacturers' instructions provided with the kit ("Leica Historesin Embedding Kit", Leica Microsystems Nussloch GmbH, Nussloch, Germany). Thin transversal sections (3.0–5.0 μm) were cut with a Leica RM2255 microtome and stained with hematoxylin-eosin and toluidine blue (see *Miranda et al., 2017*). Slides were observed and photographed under a Zeiss microscope Primo Star (AxioCam ERc5s). The slides were deposited in the collection of the Laboratory of Biology and Evolution of Cnidaria (catalog number LABEC 001, voucher deposited at the Taxonomic Collections Center of the Federal University of Minas Gerais,

**Table 1 Holotype, paratypes and additional material deposited in museums for this study.**

| Institution | Catalog number | Data |
|---|---|---|
| Smithsonian Institution, National Museum of Natural History | USNM 1522349 | 1 Holotype in 10% formalin, collected July 15, 2019 |
| Smithsonian Institution, National Museum of Natural History | USNM 1522350 | 3 Paratypes in 10% formalin, collected July 15, 2019 |
| Smithsonian Institution, National Museum of Natural History | USNM 1522351 | 5 Specimens in 95% ethanol, collected July 15, 2019 |
| Smithsonian Institution, National Museum of Natural History | USNM 1522352 | 11 Specimens in 10% formalin, collected July 3, 2019 |
| Smithsonian Institution, National Museum of Natural History | USNM 1522353 | 3 Specimens in 10% formalin, collected July 29, 2019 |
| Florida Museum at the University of Florida | UF Cnidaria 15456 | 3 Paratypes in 10% formalin, collected July 15, 2019 |
| Florida Museum at the University of Florida | UF Cnidaria 15457 | 11 Specimens in 10% formalin, collected July 3, 2019 |
| Florida Museum at the University of Florida | UF Cnidaria 15458 | 5 Specimens in 95% ethanol, collected July 15, 2019 |
| California Academy of Sciences | CASIZ 236463 | 3 Paratypes in 10% formalin, collected July 15, 2019 |
| California Academy of Sciences | CASIZ 236464 | 11 Specimens in 10% formalin, collected July 3, 2019 |
| California Academy of Sciences | CASIZ 236465 | 2 Specimens in 10% formalin, collected July 29, 2019 |
| California Academy of Sciences | CASIZ 236466 | 5 Specimens in 95% ethanol, collected July 15, 2019 |
| Royal British Columbia Museum | 022-00040-001 | 3 Paratypes in 10% formalin, collected July 15, 2019 |
| Royal British Columbia Museum | 022-00040-002 | 5 Specimens in 95% ethanol, collected July 15, 2019 |
| Royal British Columbia Museum | 022-00041-001 | 11 Specimens in 10% formalin, collected July 3, 2019 |
| Royal British Columbia Museum | 022-00042-001 | 3 Specimens in 10% formalin, collected July 29, 2019 |
| Friday Harbor Laboratories Synoptic Collection | #3140 | 11 Specimens in 10% formalin, collected July 3, 2019 |
| Friday Harbor Laboratories Synoptic Collection | #3141 | 5 Specimens in 95% ethanol, collected July 15, 2019 |
| Taxonomic Collections Center of the Federal University of Minas Gerais | UFMG-INV 2000003, Slides LABEC 001 | 1 Specimen in 10% formalin, collected May 18–19, 2015 |

Brazil, UFMG-INV 2000003; Table 1), Department of Zoology, Institute of Biological Sciences, Federal University of Minas Gerais, Brazil.

## Distribution

In addition to searching the scientific literature, the geographic range of *H. sanjuanensis* n. sp. has been determined by contacting many colleagues who work in the intertidal in British Columbia, Canada and Washington, Oregon, and California, USA, and by searching the web for images posted online and subsequently contacting and discussing the images with the photographers. Images posted on iNaturalist.org were useful in verifying some field locations (Table S1). All locations were verified with photographs or specimens. The distributional map was constructed using Q-GIS 3.28. Geographic data were georeferenced mostly by our colleagues who provided images which we examined, and occasionally from site names described in the literature (Table S1) using Google™ Earth (Pro, v7) or Google Maps using satellite view so that shoreline locations could be precisely determined.

## RESULTS AND DISCUSSION
SYSTEMATIC ACCOUNT
**Phylum Cnidaria Verrill, 1865**

**Subphylum Medusozoa Petersen, 1979**
**Class Staurozoa Marques & Collins, 2004**
**Order Stauromedusae Haeckel, 1879**
**Suborder Myostaurida** *Miranda et al. (2016a)*
**Family Haliclystidae Haeckel, 1879**
**Genus** *Haliclystus* *James-Clark, 1863*
*Haliclystus sanjuanensis* **new species**

**Synonymy.**
"*Haliclystus sanjuanensis* n. sp." in *Gellermann 1926* (pp. 80–84, Plate XX figures 20, 20a, Plate XXXVI figures 49, 50, 51) San Juan Island, Washington, USA.
"*Haliclystus stejnegeri*" in *Ricketts 1929* (p. 20).
"Stalked Jelly Fish *Haliclystus sanjuanensis*" in *Guberlet 1936*, *1949* (pp. 46, 47, 409).
"*Haliclystus*" in *Ricketts & Calvin 1939* (p. 87).
"*Haliclystus* (probably *stejnegeri*)" in *Ricketts & Calvin 1939* (p. 195) Port Townsend and Friday Harbor, Washington, USA and from Point Joe, Monterey, and Carmel Bay, California, USA.
"*Haliclystus stejnegeri*" in *Ricketts & Calvin 1939* (Figure 86 and p. 261).
"*Haliclystus*" in *Hyman 1940a* (Figure 165E and caption on p. 510, pp. 534, 535) Puget Sound (San Juan Island), Washington, USA.
"*Haliclystus*" in *Hyman 1940b* (Figure 8 caption and text on p. 292, pp. 282, 293, 295) San Juan Island, Washington, USA.
"*Haliclystus sanjuanensis*" in *Hyman 1940b* (p. 292) San Juan Island, Washington, USA.
"*Haliclystus*" in *Ricketts & Calvin 1948* (p. 87).
"*Haliclystus* (probably *stejnegeri*)" in *Ricketts & Calvin 1948* (p. 195) Port Townsend and Friday Harbor, Washington, USA and from Point Joe, Monterey, and Carmel Bay, California, USA.
"*Haliclystus stejnegeri*" in *Ricketts & Calvin 1948* (Figure 86 and p. 296).
"*Haliclystus stejnegeri*" in *MacGinitie & MacGinitie 1949* (Figure 19 on p. 120, p. 121) found in great numbers in Puget Sound and occasionally in Monterey Bay, USA.
"*Haliclystus sanjuanensis*" in *Knapp 1950*, Neptune State Park, Oregon, USA.
"*Haliclystus*" in *Ricketts, Calvin & Hedgpeth 1952* (p. 85).
"*Haliclystus* (probably *stejnegeri*)" in *Ricketts, Calvin & Hedgpeth 1952* (p. 249) Port Townsend and Friday Harbor, Washington, USA and from Point Joe, Monterey, and Carmel Bay, California, USA.
"*Haliclystus stejnegeri*" in *Ricketts, Calvin & Hedgpeth 1952* (Figure 86 on p. 284, p. 416).
"Two species of *Haliclystus* are found in the Friday Harbor region" in *Smith et al. 1954* (p. 41) San Juan Islands, Washington, USA.
"*Haliclystus auricula*" in *Smith 1955*, Field Notebook #3, June 24, 1955, Davis Bay, Lopez Island, Washington, USA.
"*Haliclystus auricula* (Rathke)" in *Gwilliam 1956* (pp. 2, 8, 17, 20, 21, 43–55, 56, 57, 61–66, 79, 80, 94–97, 110–180) San Juan Island, Washington, USA.

"*Haliclystus sanjuanensis*" in *Gwilliam 1956* (pp. 13, 14, 20) San Juan Island, Washington, USA.

"*Haliclystus auricula*" in *Anonymous 1958* (p. 2) San Juan County, Washington, USA.

"*Haliclystus auricula* (Rathke)" in *Gwilliam 1960* (pp. 454–473) San Juan Island, Washington, USA.

"*Haliclystus auricula* (Rathke, 1806)" (in part) in *Kramp 1961* (pp. 292–293).

"*Haliclystus* sp." in *McLean 1962* (p. 106) scarce on rock walls along the open coast of the Monterey Peninsula, nine miles south of Carmel near Granite Creek, California, USA.

"*Haliclystus*" in *Ricketts, Calvin & Hedgpeth 1962* (p. 85).

"*Haliclystus* (probably *stejnegeri*)" in *Ricketts, Calvin & Hedgpeth 1962* (p. 249) Port Townsend and Friday Harbor, Washington, USA and from Point Joe, Monterey, and Carmel Bay, California, USA.

"*Haliclystus stejnegeri*" in *Ricketts, Calvin & Hedgpeth 1962* (Figure 86 on p. 284, p. 416).

"*Haliclystus* sp." in *MacGinitie & MacGinitie 1968* (Figure 19 on p. 120, p. 123) abundant in Puget Sound waters, USA.

"*Haliclystus*" in *Ricketts, Calvin & Hedgpeth 1968* (p. 130).

"*Haliclystus* (probably *H. stejnegeri*)" in *Ricketts, Calvin & Hedgpeth 1968* (pp. 299–300) Port Townsend and Friday Harbor, Washington, USA and from Point Joe, Monterey, and Carmel Bay, California, USA.

"*Haliclystus stejnegeri*" in *Ricketts, Calvin & Hedgpeth 1968* (Figure 221 on p. 299, p. 462).

"*Haliclystus*" in *Westfall 1968* (p. 803), San Juan Island, Washington, USA.

"*Haliclystus auricula*" in *Westfall 1968* (p. 804), San Juan Island, Washington, USA.

"Two species of *Haliclystus* are found in the Friday Harbor region" in *Smith 1970* (p. 41) San Juan Islands, Washington, USA.

"*Haliclystus auricula*" in *Westfall, Yamataka & Enos 1970* (p. 545), San Juan Island, Washington, USA.

"*Haliclystus auricula*" in *Westfall 1973* (pp. 239–243), San Juan Island, Washington, USA.

"*Haliclystus auricula*" in *Kozloff 1973* (Figure 206 and text, p. 247).

"*Haliclystus auricula*" in the Bodega Marine Lab Synoptic Collection, collected in 1973 at Shell Beach, Sonoma County, California, USA.

"*Haliclystus auricula* (Rathke, 1806)" in *Kozloff 1974* (pp. 22, 219).

"*Haliclystus auricula*" in *Smith 1975* (Figure 21 on p. 88, p. 96).

"21, *Haliclystus*, from False Bay near Friday Harbor, Wash." in *Smith 1975* (caption on p. 89).

"*Haliclystus auricula* (Rathke, 1806) (= *H. sanjuanensis* of several authors)" in *Smith 1975* (p. 96) Friday Harbor, Washington, USA.

"*Haliclystus auricula*" in *Smith 1976* (Figure 121 on p. 115) Pacific Northwest.

"*Haliclystus stejnegeri*" in *Otto 1976* (pp. 319–322, 328) San Juan Island, Washington, USA.

"*Haliclystus stejnegeri*" in *Singla 1976* (pp. 533–539) on eelgrass, Victoria, British Columbia, Canada.

"*Haliclystus auricula*" in *Flora & Fairbanks 1977* (Figure 48 on p. 56) on kelp and eelgrass in protected bays in Puget Sound, USA.

"*Haliclystus stejnegeri*" in *Otto 1978* (p. 22) San Juan Island, Washington, USA.

"*Haliclystus auricula*" in *Snively 1978* (p. 162).

**Not** "*Haliclystus auricula*" = *Manania handi* in *Snively 1978* (Figure 79 on p. 43).

"*Haliclystus*" in *Brusca & Brusca 1978* (pp. 52–54, Figure 24 on p. 53) in Northern California, USA attached to seaweeds in quiet tidepools, or to eelgrass in Humboldt Bay, USA.

"*Haliclystus auricula*" in *Caine 1980* (p. 162) San Juan Island, Washington, USA.

"*Haliclystus* sp." in *Barr & Barr 1983* (Figure 32 on p. 9, pp. 81–82) Yakobi Island, Southeast Alaska, USA.

"*Haliclystus stejnegeri* and *H. auricula*" in *Phillips 1984* (p. 40) Pacific Northwest.

"*Haliclystus auricula* (Rathke, 1806)", in part, in *Austin 1985* (p. 70).

"*Haliclystus steinegeri Kishinouye, 1899*", in part, in *Austin 1985* (p. 70).

"*Haliclystus*" in *Ricketts et al. 1985* (pp. 154–155) common "in the north" and rare to occasional at Monterey, USA.

"*Haliclystus auricula*" in *Ricketts et al. 1985* (pp. 343–344, Figure 267 on p. 344, p. 513) on eelgrass in Puget Sound and northern California, USA.

"*H. stejnegeri*" in *Ricketts et al. 1985* (p. 513).

"*H. sanjuanensis*" = *Haliclystus auricula* (Rathke, 1806) in *Hirano 1986* (p. 183).

"*Haliclystus*" in *Westfall 1987* (pp. 9, 11, 12), San Juan Island, Washington, USA.

"*Haliclystus stejnegeri Kishinouye, 1899*" in *Mills 1987* (p. 67).

"*Haliclystus stejnegeri* Kishinouye (= *H. sanjuanensis* and *H. auricula*, E. Kozloff personal communication to C.E.M. *c.* 1984)" in *Strathmann 1987* (p. 79) San Juan Island, Washington, USA.

"*Manania* (?)" DP Abbott in *Hilgard 1987* (p. 22) July 2, 1974, 15–20 feet deep on *Plocamium*, off Malpaso Creek, Monterey County, California, USA.

"*Haliclystus octoradiatus* (Lamarck, 1816)" in *Larson 1990* (pp. 546–550, Figure 1A on p. 551) Alaska to central California, USA.

"*Haliclystus stejnegeri Kishinouye, 1899*" in *Cairns et al. 1991* (p. 11 (in part), Figure 1 on p. 77).

"*Haliclystus octoradiatus*" in *Eckelbarger & Larson 1993* (pp. 225–236) Victoria, British Columbia, Canada.

"*Haliclystus salpinx*" in *Eckelbarger 1994* (p. 16, Figure 2.3.d on p. 19, pp. 20, 26) Victoria, British Columbia, Canada.

"*Haliclystus*" in *Kozloff 1996* (pp. 322, 323).

"*Haliclystus stejnegeri*" in *Kozloff 1996* (p. 323, Figure 345 on p. 324).

"*Haliclystus stejnegeri Kishinouye, 1899*" in *Mills 1996* (pp. 67, 489).

"*H. sanjuanensis* Hyman, 1940" in *Hirano 1997* (pp. 247, 249–251).

"*Haliclystus stejnegeri*" in *Heeger 1998* (p. 74).

"*Haliclystus stejnegeri*" in *O'Clair & O'Clair 1998* (Figure 13 on p. 15, pp. 15–16) Bertha Bay, west side of Chichagof Island, Southeast Alaska, USA.

"*Haliclystus sanjuanensis*" in *Wrobel & Mills 2003* (p. 49).

"*Haliclystus* "sanjuanensis" *Gellermann, 1926* or Hyman, 1940 or *Hirano, 1997* or undescribed?" in *Mills 1999–present* accompanied by C.E. Mills photograph of two stauromedusae on one piece of eelgrass (accessed March 2023).

"*Haliclystus sanjuanensis*" in *Collins 2000* (pp. 10, 11, 14–16).

"*Haliclystus* sp. undescribed "*sanjuanensis*"" in *Cairns et al. 2002* (pp. 9, 47).

"*Haliclystus stejnegeri*" in *Cairns et al. 2002* (Figure 3 on p. 109).

"*Haliclystus sanjuanensis*" in *Collins 2002* (p. 420).

"*Haliclystus sanjuanensis* Hyman, 1940" in *Zagal 2004* (table 2 on p. 339).

"*Haliclystus sanjuanensis* Hyman, 1940" in *Collins & Daly 2005* (pp. 221, 222, 225, 226, 229).

"*Haliclystus sanjuanensis*" in *Collins et al. 2006a* (p. 3).

"*Haliclystus sanjuanensis*" in *Collins et al. 2006b* (Figure 2 on p. 101, Figure 3 on p. 102, Figure 4 on p. 103, Figure 6 on p. 106, p. 115).

"Oval-anchored stalked jelly *Haliclystus stejnegeri*" in *Lamb & Hanby 2005 & 2015* (p. 111 photo and text) intertidal to 10 m, Sekiu, Olympic Peninsula, Washington, USA.

"*Haliclystus* sp." in *Mills & Hirano 2007* (Figure 1 on p. 541).

"*Haliclystus* sp. (undescribed species)" in *Mills & Larson 2007* (p. 170).

"*Haliclystus* sp. ("*sanjuanensis*")" in *Mills & Larson 2007* (Plate 63A on p. 171).

"*Haliclystus* sp. (=*H.* "*sanjuanensis*", nomen nudum)" in *Mills & Larson 2007* (p. 172).

"*Haliclystus* "*sanjuanensis*" (*nomen nudum*)" in *Miranda, Morandini & Marques 2009* (pp. 1507, 1508, 1513–1515).

"*Haliclystus* "*sanjuanensis*" (*nomen nudum*)" in *Miranda, Collins & Marques 2010* (pp. 3–5, 7).

"*H. sanjuanensis*" in *Miranda, Collins & Marques 2010* (p. 4).

"*Haliclystus* "*sanjuanensis*" nomen nudum" in *Kahn et al. 2010* (pp. 49, 52, 54–57).

"Oval-anchored stalked jelly (*Haliclystus stejnegeri*)" in *Brewer et al. 2011* (p. 40 photo and text) Cape Prominence, Unalaska Island in the Aleutians, USA.

"*Haliclystus* "*sanjuanensis*"" in *Kayal et al. 2012* (pp. 2–4, 11).

"*Haliclystus sanjuanensis*" in *Kayal et al. 2013* (Figure 2 on p. 5, table 3 on p. 13).

"*H.* "*sanjuanensis*"" in *Miranda, Collins & Marques 2013* (p. 1381).

"*Haliclystus sanjuanensis*" in *Zapata et al. 2015* (pp. 5, 9, Figure 4 on p. 7) San Juan Island, Washington, USA.

"*Haliclystus* 'sanjuanensis' nomen nudum" in *Westlake 2015* (185 pp.) Victoria, British Columbia, Canada.

"*H.* "*sanjuanensis*" nomen nudum" in *Miranda et al. 2016a* (pp. 6, 13–15, 17, 19, 37).

"*H.* "*sanjuanensis*" (*nomen nudum*)" in *Miranda et al. 2016b* (pp. 3, 72, 73).

"*Haliclystus sanjuanensis*" in *Li, Sung & Ho 2016* (pp. 113–114).

"*Haliclystus* 'sanjuanensis' nomen nudum" in *Westlake & Page 2017* (pp. 29–49) Victoria, British Columbia, Canada.

"*Haliclystus* "*sanjuanensis*"" in *Kayal et al. 2018* (pp. 4, 13, 15, Figure 1b on p. 2, Figure 5 on p. 7, Figure 6 on p. 9) from Cattle Point, San Juan Island, Washington, USA.

"*H.* "*sanjuanensis*" nomen nudum" in *Miranda et al. 2018* (pp. 1697, 1699, 1703, 1706, 1707, 1709).

"*Haliclystus* sp." in *Jensen, Gotshall & Flores Miller 2018* (p. 28) British Columbia, Canada to central California, USA.

"*Haliclystus sanjuanensis*" in *Nicosia et al. 2018* (Table 2 on p. 5, Figure 6 on p. 13).

"*Haliclystus "sanjuanensis""* and "*Haliclystus sanjuanensis*" in *Holst & Laakmann 2019* (Table 1 on p. 1062, Figure 2 on p. 1064).

"*Haliclystus "sanjuanensis""* in *Holst, Heins & Laakmann 2019* (Figure 6 on p. 1787, Table 4 on p. 1788, pp. 1789, 1793, 1794).

"*Haliclystus "sanjuanensis"* (*nomen nudum*)" in *Miranda & Collins 2019* (p. 8).

"*Haliclystus sanjuanensis*" in *Koch & Grimmelikhuijzen 2019* (Table 1 on p. 3, Table 3 on p. 5, Figure 3 on p. 9, Figure 4 on p. 10).

"*Haliclystus sanjuanensis*" in *Lewandowska, Hazan & Moran 2020* (p. 11).

"*Haliclystus sanjuanensis*" in *Gornik et al. 2021* (p. 1755).

"*Haliclystus sanjuanensis*" in *Hartigan et al. 2021* (Figure 4 on p. 11).

"*Haliclystus "sanjuanensis"* (*nomen nudum*)" in *Holst et al. 2021* (p. 249).

"*Haliclystus sanjuanensis*" in *Adriansyah et al. 2022* (Figure 8A on p. 265, Table 2 on p. 269).

"*Haliclystus sanjuanensis*" in *Santander et al. 2022* (Figure 1C on p. 2).

"*Haliclystus sanjuanensis*" in *Vöcking et al. 2022* (p. 11).

"*Haliclystus sanjuanensis*" in *Zhang & Jacobs 2022* (Table 1 on p. 5, Figure 4 on p. 8).

"*Haliclystus sanjuanensis*" in *Steinworth, Martindale & Ryan 2023* (Table 2 and text on p. 3, p. 9).

**We remain unsure whether the following records refer to *H. sanjuanensis*:**

"*Haliclystus stejnegeri* Kishinouye" in *Bigelow (1920)* (p. 12H and Plate 2 Figure 4 on p. 20H) Port Clarence, Alaska, USA, 2–3 fathoms. There is insufficient information to distinguish if this was *H. stejnegeri* or *H. sanjuanensis*.

"*Haliclystus*" in 1946 field notes by Ed Ricketts (*Transcript of the 1946 trip to West Vancouver Island and to the Queen Charlotte Islands*), Massett, Queen Charlotte Islands, Canada, June 27, 1946, and July 2, 1946, "lots of *Haliclystus*" on eelgrass in two locations near Massett in the very low intertidal, in *Steinbeck, Ricketts & Hedgpeth (1978)* (pp. 115, 118), and *Rodger (2006)* (pp. 306, 310).

"*Haliclystus auricula* (Rathke, 1806)" in *Nuclear Regulatory Commission (2001)*, Pacific Gas & Electric Environmental Report, ~2001, table 2.3–5, Sheet 7 of 23, p. 173 of 202, from coastline near Diablo Canyon, California, USA, (confirmed by personal communication from David Behrens, who did much of the collecting and identifying for that report, but since he could not locate a photograph of the specimen for us in 2019, we cannot be positive of this material, which would otherwise represent one of the southernmost collection locations of *H. sanjuanensis* n. sp.).

"*Haliclystus stejnegeri*" Royal British Columbia Museum (RBCM) #015-00395-001. Single animal collected from Egeria Bay, Langara Island, Haida Gwaii, British Columbia, Canada, May 22, 2015 by Heidi Gartner, Melissa Frey, Gavin Hanke, Tommy Norgard, Leslie Barton and Vanessa Hodes is too young to determine whether it is *H. sanjuanensis* n. sp. or *H. stejnegeri*. Claudia Mills notes upon inspection of the specimen, April 26, 2019.

**Holotype.** Smithsonian Institution, National Museum of Natural History, USNM 1522349 (Table 1). Northeast Pacific, collected July 15, 2019 at Cattle Point, San Juan Island, San Juan County, Washington, USA (48.452131° N, 122.961750° W) on a blade of algae at

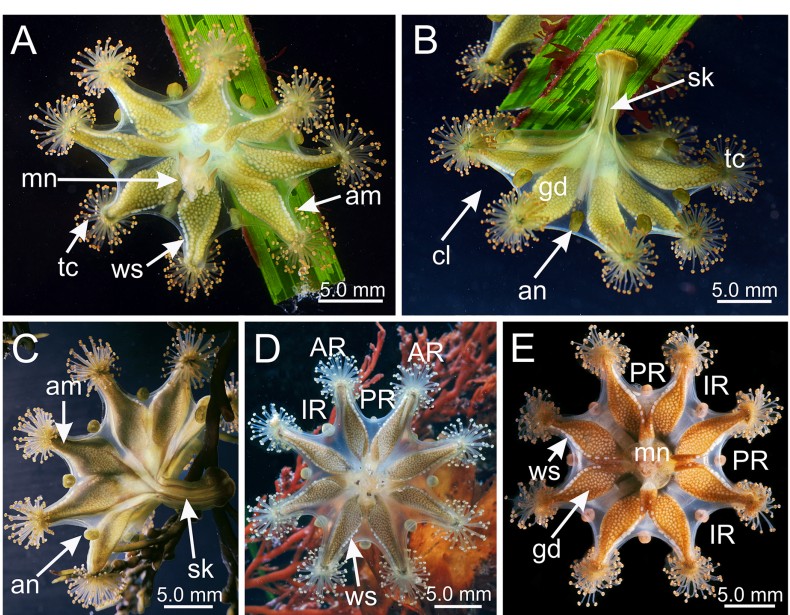

**Figure 1** *Haliclystus sanjuanensis* **n. sp. in the field.** (A, D and E) Oral view; (B and C) lateral view. (A and B) images courtesy of Gustav Paulay, collected from Cattle Point, San Juan Island, USA (type locality); (C–E) images courtesy of Ron Larson from (C) Saxe Point, Victoria, British Columbia, Canada; (D) and (E) Seal Rock, Oregon, USA. am, arm; an, anchor; AR, adradii; cl, calyx; gd, gonad; IR, interradii; mn, manubrium; PR, perradii; sk, stalk; tc, secondary tentacles; ws, white nematocyst spot.

approximately 0.5 m deep. Preserved in 10% formalin in seawater. Collector Claudia E. Mills.

**Other material examined.** Paratypes and additional specimens (Table 1) were collected July 3, July 15 and July 29, 2019, on low tides at Cattle Point, San Juan Island, San Juan County, Washington, USA (48.452131° N, 122.961750° W) on red or green algae or surfgrass *Phyllospadix* at approximately 0.5 m deep, collector Claudia E. Mills. The animals were preserved in 10% formalin in seawater or preserved directly into 95% ethanol. Both formalin and ethanol preserved paratypes and additional specimens were deposited at the Smithsonian Institution, National Museum of Natural History in Washington, DC; the Florida Museum at the University of Florida in Gainesville; the Royal British Columbia Museum in Victoria, British Columbia; the California Academy of Sciences in San Francisco, California; and the Synoptic Collection of the University of Washington Friday Harbor Laboratories in Friday Harbor, Washington (Table 1).

**Further evidence.** Photographs (mainly from https://www.inaturalist.org/), specimens in public and particular collections, and personal communications, not otherwise published, South to North, are available in Table S1.

**Diagnosis.** *Haliclystus* with broadly conical calyx, wider than high (Figs. 1 and 2). Calyx well demarcated from the stalk, where the profile goes from conical to cylindrical (Figs. 1 and 2). Stalk length variable, from as long as the calyx is tall to about 1/2 length of calyx. Gonads extending from base of the calyx to tips of each of eight arms (Figs. 1 and 2); 200 to 300 tightly-packed gonadal vesicles comprising each gonad, with 10 to 22 abreast at the

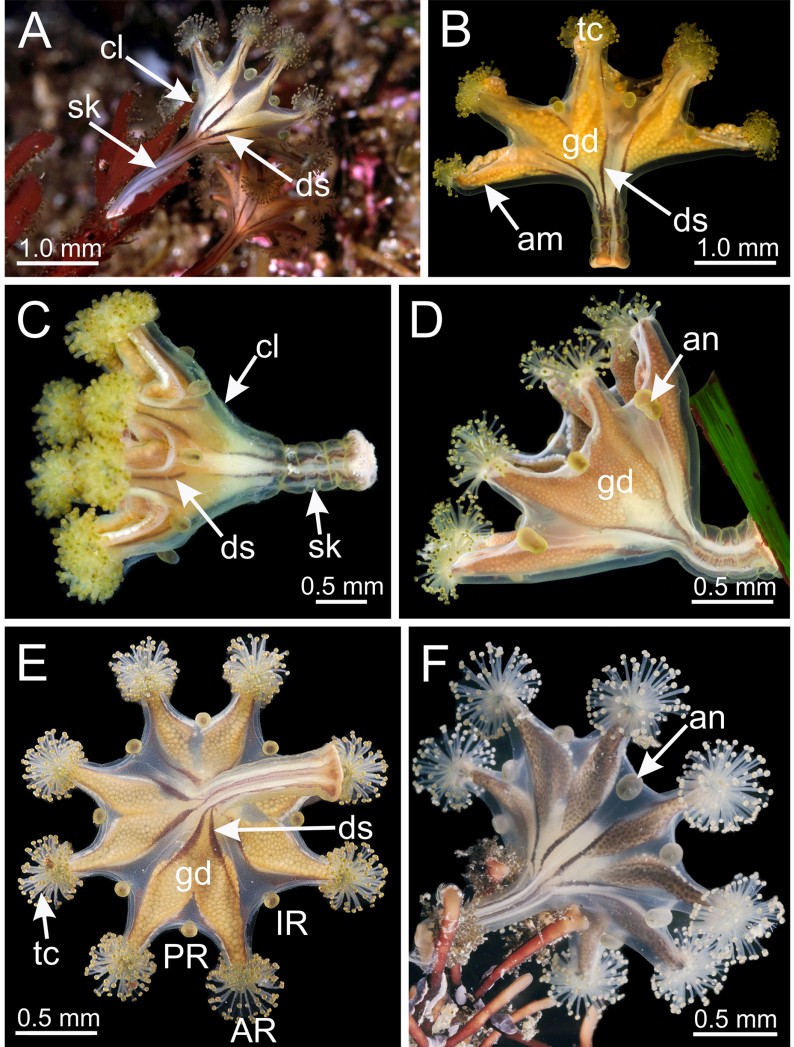

**Figure 2 Dark stripes pattern in _Haliclystus sanjuanensis_ n. sp. (see also Table S2).** (A–D and F) Lateral view; (E) aboral view. (A) Beresford Island, British Columbia, Canada, image courtesy of Neil McDaniel; (B–D) Calvert Island, British Columbia, Canada, images courtesy of Gustav Paulay; (E and F) Seal Rock, Oregon, USA, images courtesy of Ron Larson. am, arm; an, anchor; AR, adradii; cl, calyx; ds, dark stripes; gd, gonad; IR, interradii; PR, perradii; sk, stalk; tc, secondary tentacles.

widest (Figs. 1 and 2). The gonads are broadly leaf-shaped, filling much of the available subumbrellar surface, and slightly folded over upon themselves in larger animals; the proximal half of each gonad may be turned over upon itself in the perradii and the distal half turned over in a similar manner in the interradii (Figs. 1 and 2). Each arm tipped with a cluster of usually up to 130 (but may be more than 200 in larger specimens) capitate secondary tentacles (Figs. 1–3). Eight anchors along the calyx margin on interradial and perradial axes, alternating with the eight adradial arms—the anchors are roughly oval in outline and vary from fairly spherical to elongate, sometimes with a remnant of the primary tentacle in the center, usually disappearing in older animals (Figs. 1, 2, 3A–3E). The eight marginal notches between arms are all approximately equal in size (Figs. 1 and

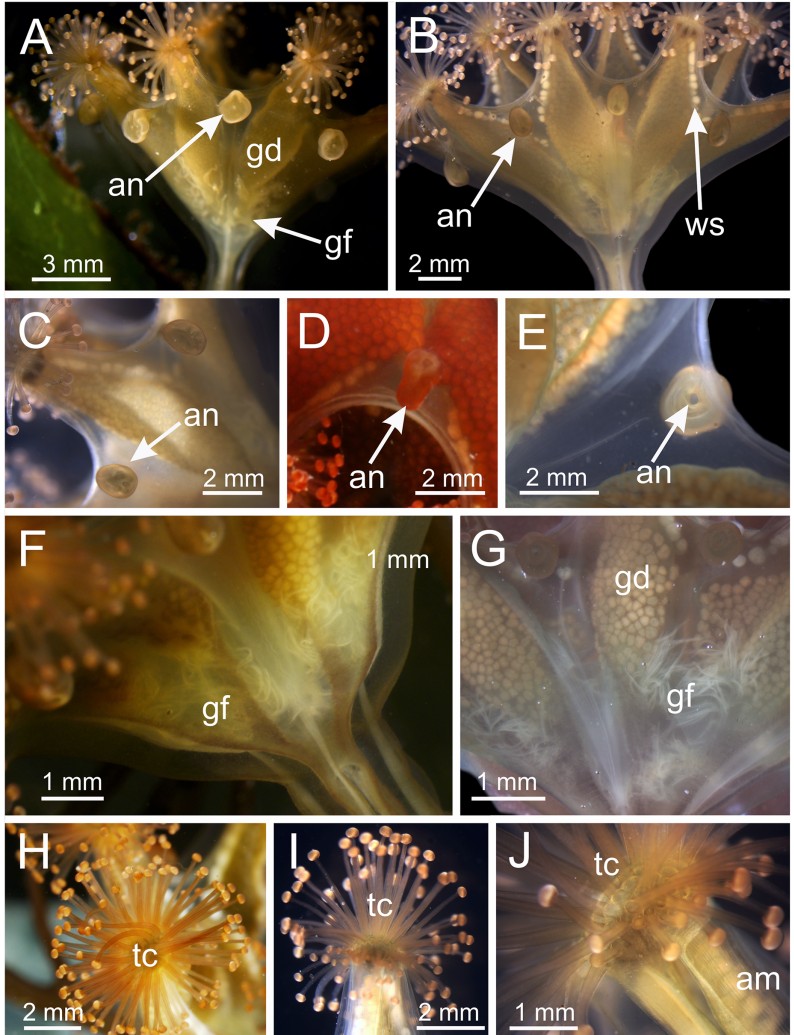

**Figure 3 Calyx of *Haliclystus sanjuanensis* n. sp., with anchors, gastric filaments, and secondary tentacles.** (A) Lateral view of a juvenile stauromedusa, with anchors with a remnant of primary tentacle; (B–D) lateral view of adult stauromedusae, with more developed anchors with smaller remnant of primary tentacles; (E) connection of anchor with calyx; (F–G) gastric filaments inside gastrovascular cavity; (H–J) secondary tentacles at arms tips. an, anchor; gd, gonad; gf, gastric filaments; tc, secondary tentacles; ws, white nematocyst spot.

2). Bright-white nematocyst spots about the same size as the gonadal vesicles (Figs. 4E–4G), closely grouped in a single row along the distal perradial edges of the gonads from the tips of the arms to the widest point of the gonads or a little further down, but also rarely seen along the perradial calyx margin (Figs. 1–4). Often, but not always, with four dark pigment stripes running up the perradii of the stalk and bifurcating (into eight stripes) at the stalk-calyx junction, extending halfway up the calyx on the exumbrella along the interradial edge of each gonad, and sometimes also with lighter stripes running along both upper edges of the gonads near the calyx margin (Fig. 2).

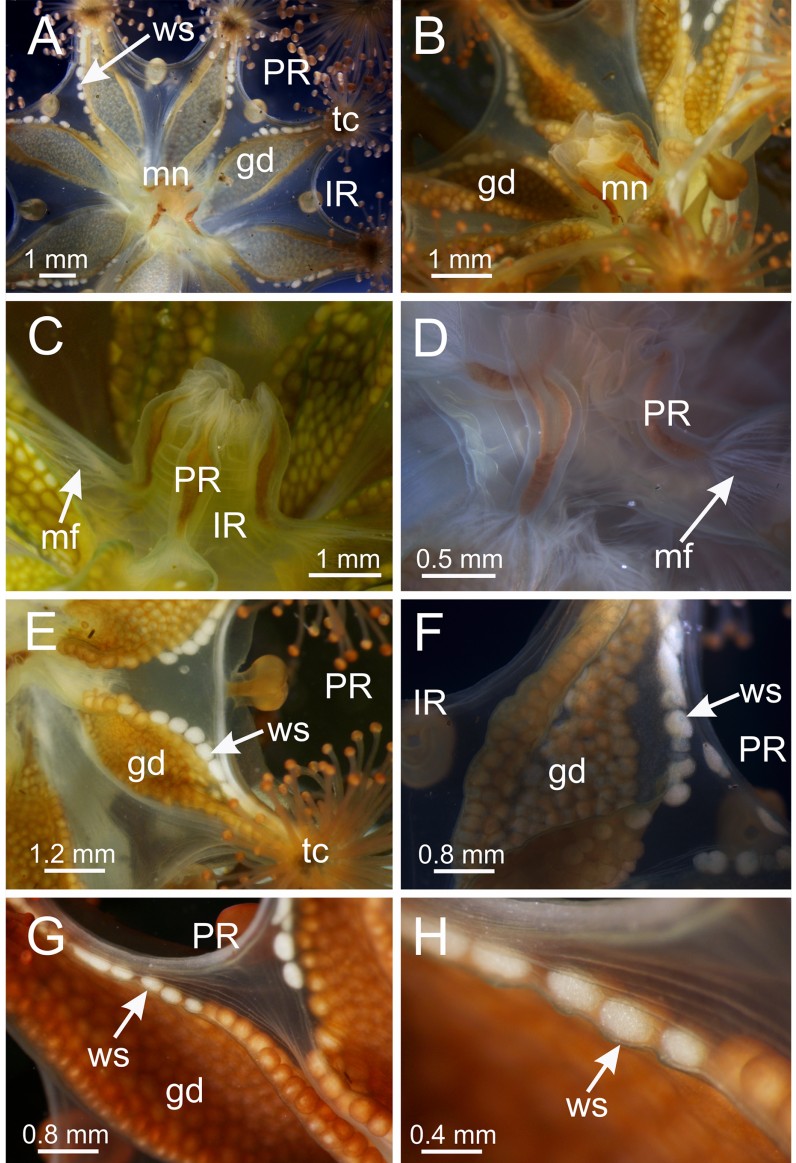

**Figure 4 Calyx of *Haliclystus sanjuanensis* n. sp. with manubrium and white nematocyst spots.** (A and B) General oral view; (C and D) detail of manubrium, with perradial muscular fibers; (E–G) general view of the pattern of distribution of white nematocyst spots; (H) detail of white nematocyst spots. gd, gonad; IR, interradii; mf, muscular fibers; mn, manubrium; PR, perradii; tc, secondary tentacles; ws, white nematocyst spot.

**Description.** (General description using *Gellermann, 1926*; *Gwilliam, 1956*; *Westlake, 2015*; photographs by the authors and available online, and observation of living and preserved individuals).

Calyx broadly conical, with eight arms; arms sometimes recurved toward the stalk in life; calyx usually up to about 30 mm wide (rarely to 40 mm wide) and clearly demarcated from the stalk, wider than high (Figs. 1 and 2). Stalk narrow and varies from about 1/2 the length of calyx to as long as the calyx is tall, rarely a little longer than the calyx is tall (Figs. 1, 2 and 5). Well-developed basal pedal disc at base of stalk (Figs. 5B and 5C). Stalk

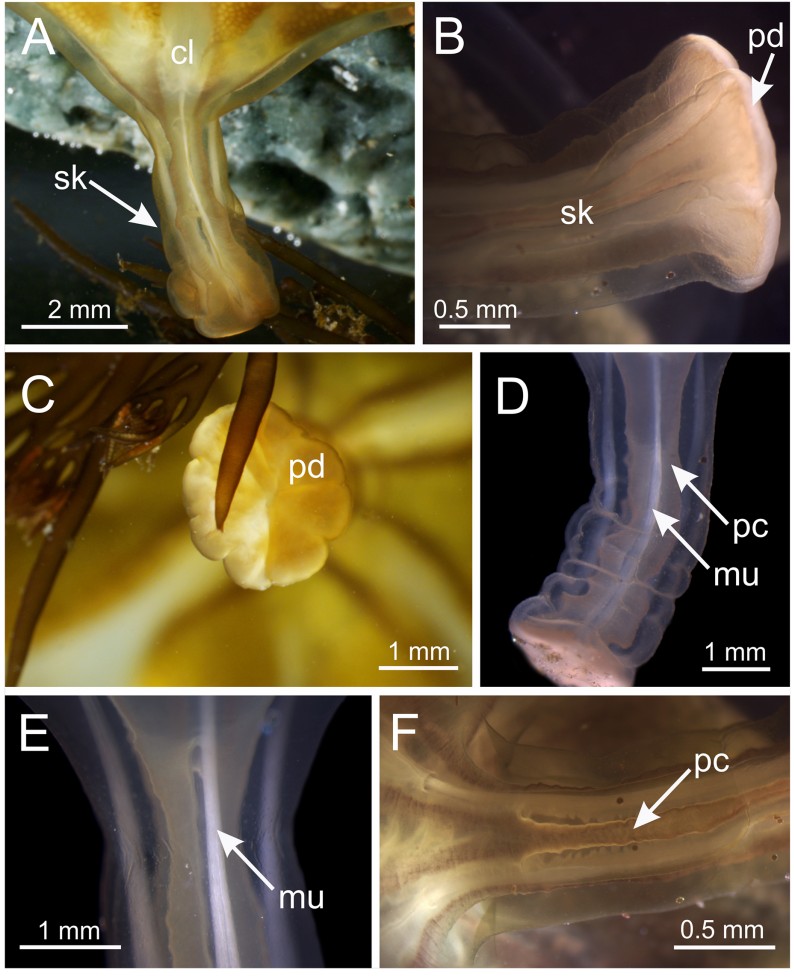

**Figure 5 Stalk of *Haliclystus sanjuanensis* n. sp.** (A) Lateral view of stalk-calyx connection; (B) lateral view of stalk, with pedal disc; (C) basal view of pedal disc; (D) lateral view of stalk, with four interradial longitudinal muscles and four perradial chambers; (E and F) detail of lateral view of stalk, with four interradial longitudinal muscles and with four perradial chambers. cl, calyx; mu, interradial longitudinal muscle; pc, perradial chamber; pd, pedal disc; sk, stalk.

quadratic in cross section, with four usually darkly-pigmented perradial gastric chambers (delimited by gastrodermis) and four interradial longitudinal muscle bands, embedded in mesoglea (Figs. 5 and 6A). Perradial chambers fusing into one single chamber at stalk-calyx connection, with four interradial septa associated with the four interradial longitudinal muscle bands (Figs. 6B, 6C and 7). Four infundibula funnel-shaped with blind end, delimited by epidermis, deeply developed down to base of calyx, widening apically, with broad apertures on subumbrella (Fig. 7). Gastrovascular cavity without claustrum. Numerous gastric filaments in gastrovascular cavity (Figs. 3F and 3G). At base of infundibula, the four interradial longitudinal muscle bands of the stalk divide into eight bands (Figs. 6C–6G, 7D and 7E), continuing up through the calyx along interradii, reaching out into the eight adradial arms. Gastrovascular cavity divided into four perradial gastric pockets at the manubrium region (Figs. 6F and 6G), where gonadal vesicles are located in mature individuals. Four perradial gastric pockets bifurcating distally to form

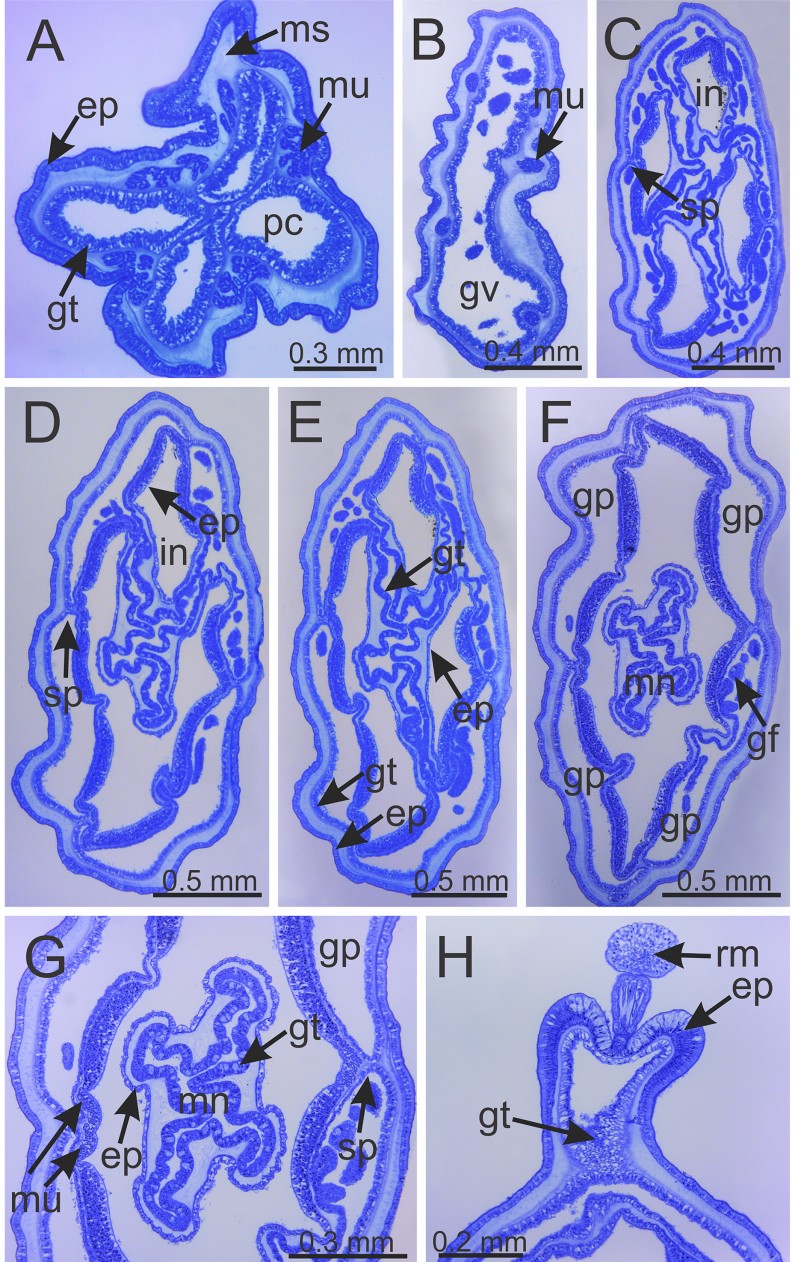

**Figure 6 Internal anatomy of stalk and calyx of a juvenile (immature)** *Haliclystus sanjuanensis* **n. sp.**
(A) stalk, with four perradial chambers and four interradial longitudinal muscles; (B) stalk-calyx connection, with fused gastric chambers into one main gastrovascular cavity; (C–E) base of manubrium region, with delimitation of four perradial gastric pockets; (F) manubrium region, with central manubrium and four perradial gastric pockets delimited; (G) detail of manubrium, with four corners; (H) anchor of a juvenile specimen, with remnant of primary tentacle. (A–G) cross-sections; (H) longitudinal section. ep, epidermis; gf, gastric filament; gp, perradial gastric pocket; gt, gastrodermis; gv, gastrovascular cavity; in, infundibulum; mn, manubrium; ms, mesoglea; mu, interradial longitudinal muscle; pc, perradial gastric chamber; rm, remnant of primary tentacle; sp, septum.

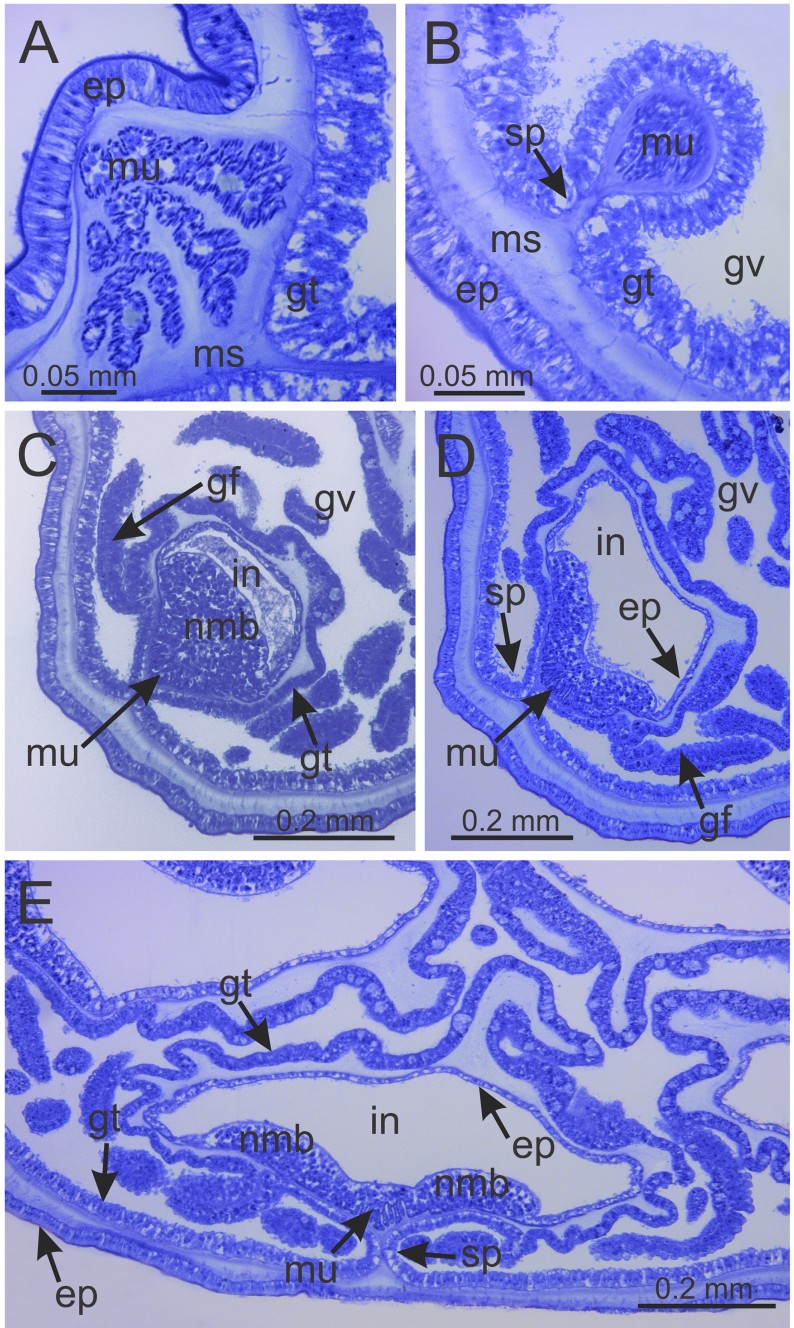

**Figure 7 Interradial longitudinal muscle and infundibula of a juvenile (immature)** *Haliclystus sanjuanensis* **n. sp.** (A) Interradial longitudinal muscle in stalk; (B) Interradial longitudinal muscle at the stalk-calyx connection; (C) Interradial longitudinal muscle and infundibulum at calyx base, with a "group" of nematoblasts; (D) wider infundibulum and compressed longitudinal muscle; (E) even wider infundibulum and more compressed longitudinal muscle, at base of manubrium region, with two "groups" of nematoblasts. (A–E) cross-sections. ep, epidermis; gf, gastric filament; gt, gastrodermis; gv, gastrovascular cavity; in, infundibulum; ms, mesoglea; mu, interradial longitudinal muscle; nmb, nematoblasts; sp, septum.

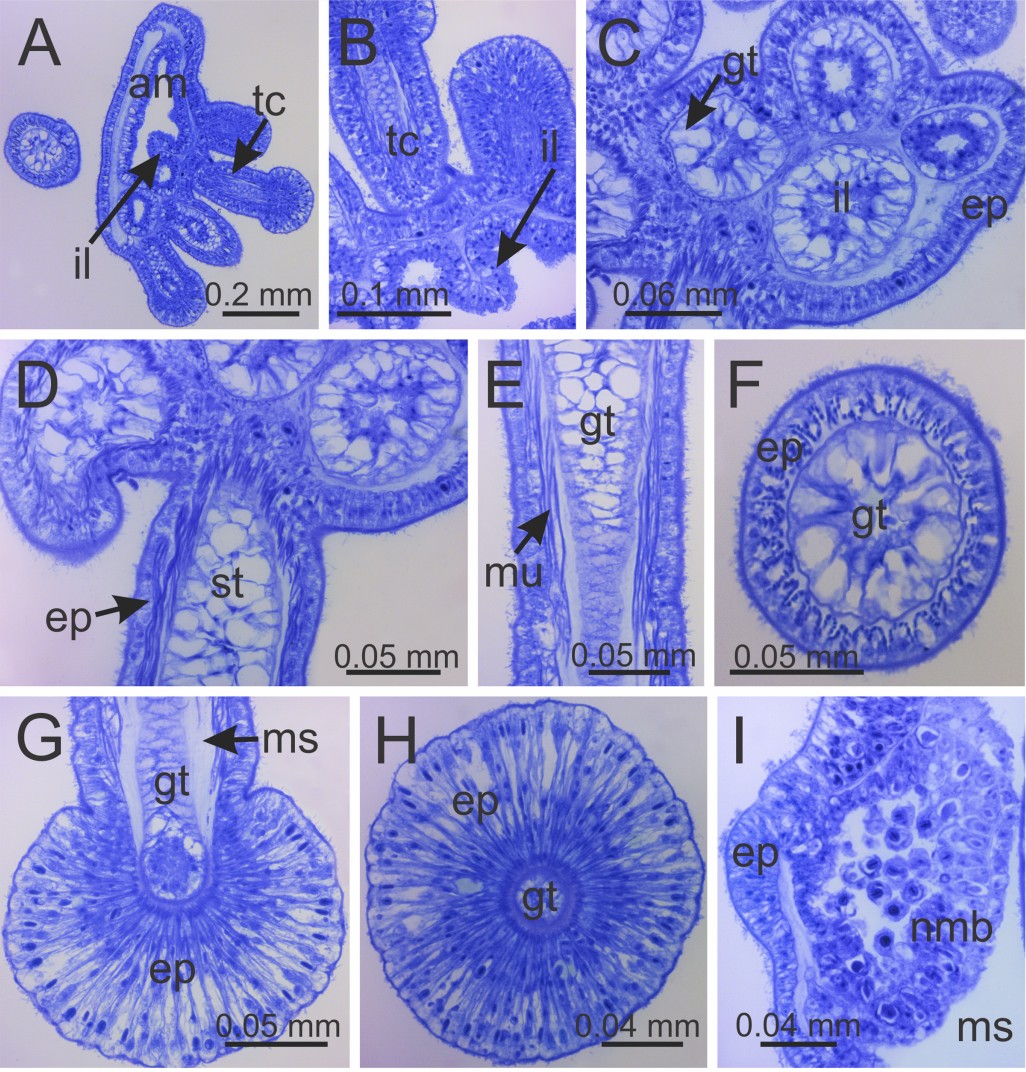

**Figure 8 Internal anatomy of arm tips, secondary tentacles, and white nematocyst spots of a juvenile (immature)** *Haliclystus sanjuanensis* **n. sp.** (A) Tip of arm, showing intertentacular lobules and secondary tentacles; (B and C) detail of intertentacular lobules; (D and E) stem of secondary tentacles, with longitudinal muscle; (F) stem of secondary tentacle; (G and H) knob of secondary tentacle; (I) white nematocyst spot. (A, B, D, E, G and I) longitudinal sections; (C, F and H) cross-sections. am, arm; ep, epidermis; gt, gastrodermis; il, intertentacular lobules; ms, mesoglea; mu, longitudinal muscle; nmb, nematoblasts; st, stem; tc, secondary tentacles.

eight adradial branches, each of which is running into each arm. The manubrium is short and quadrangular, with dark-pigmented perradial corners, and with the four oral lobes thin and folded (Figs. 4A–4D, 6F and 6G).

Eight adradial arms, all alike, and notches between arms are all approximately equal in size (Figs. 1 and 2). However, arms occasionally exhibit some perradial pairing, with the perradial notches between arms sometimes a little narrower than the interradial notches. Tip of each arm with a cluster of up to 130 (or more than 200 in larger specimens) knobbed (capitate) secondary tentacles; all tentacles are short, from 1 to 3 mm long (Figs. 1–3 and 8A–8H).

Intertentacular lobules at the internal tip of the arms (at the base of tentacular clusters) (Figs. 8A–8C). Each secondary tentacle in the arms is ensheathed with longitudinal muscle fibers (Figs. 8D–8F). Four groups of perradial muscular fibers extend from the manubrium lips (Figs. 4C and 4D) down to the four perradial corners of the manubrium. The coronal muscle lies beneath the subumbrellar surface at the periphery of the calyx and is comprised of four interradial and four perradial segments.

Eight anchors on the exumbrella, near the margin, one between each pair of arms in the interradii and perradii outside of the low-point of each notch (Figs. 1, 2 and 3A–3E). The anchors are irregularly oval in outline and about half as long as the diameter of the stalk, but quite variable in shape and size between animals. The primary tentacle (the developmental origin of the anchor) capitulum usually disappears in older specimens, but is present in developing animals to a greater or lesser extent depending on animal size (Figs. 3A–3E and 6H). The anchors may be fairly flat in profile or rather elongate, but they are never flaring or trumpet-shaped with thickened or rolled edges (Figs. 1, 2 and 3A–3E); the cavity of each anchor communicates with the general body cavity of the animal by means of a very short canal through the anchor peduncle (Fig. 3E).

The eight adradial gonads extend the full length of the eight arms, are very broad in the middle, filling much of the available subumbrellar surface, and in large specimens slightly folding over on themselves, and narrower at both ends (Figs. 1, 2 and 4E–4G). The gonads in the largest animals are composed of 200 to 300 tightly-packed gonadal vesicles that are roughly spherical, but may be so tightly packed together that their shape may be modified (Figs. 1, 2 and 4E–4G). The gonadal vesicles are not arranged in rows; there are 10 to 22 vesicles abreast at the broadest part of each gonad (usually 12–16) (Figs. 1 and 2). In the largest animals, the proximal half of each gonad is turned over upon itself in the perradii and the distal half is turned over in a similar manner in the interradii (Figs. 1A and 1E). The gonads are separated in the interradii, but are closer at their bases in the perradii, making four pairs of gonadal bands (Figs. 1 and 2).

The exumbrellar surface is finely granulated with numerous, scattered nematocyst clusters (Fig. 9L). The subumbrella is marked with distinctive bright-white nematocyst spots that are about the same size as the gonadal vesicles (Figs. 4E–4G)—these white spots are arranged in the perradii, mostly closely grouped in a single row along the distal perradial edges of the gonads from the tips of the arms to the widest point of the gonads or a little further down, but also occasionally located along the calyx margin (perradial center) in large specimens (Figs. 1, 4E–4H and 8I). The subumbrellar epidermis is raised in slight bulges over each white nematocyst spot (Figs. 4H and 8I).

Overall color in life is highly variable, but each individual is of a single color, from brown or beige to orangey red to peach, rusty or wine red, or greenish, usually cryptically matching the color of its eelgrass or algal substrate (Figs. 1 and 2); young (small) animals are less pigmented. Gonads and anchors golden brownish to green to orange to wine red, bright-white nematocyst spots along perradial edges of gonads and (rarely) subumbrellar margin; tentacles greenish to golden yellow to reddish or pink (Figs. 1 and 2). Often, but not always, with four dark brown to orange longitudinal stripes of varying intensity on the perradial corners of the manubrium (Figs. 4A–4D) and on the perradii of the stalk,

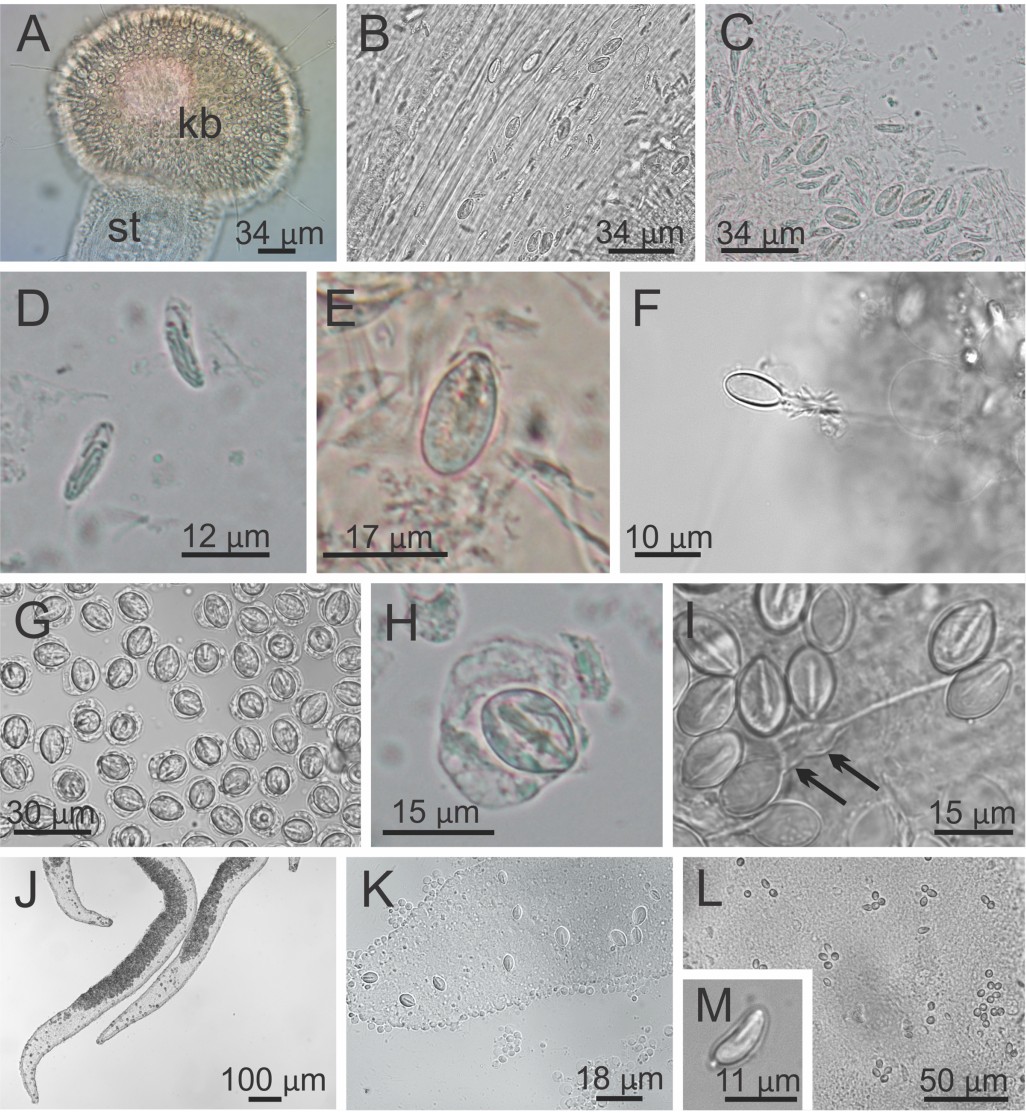

**Figure 9 Nematocysts of *Haliclystus sanjuanensis* n. sp.** (A) Knob of a secondary tentacle, with nematocysts covering the capitate tip; (B) stem of a secondary tentacle, with migrating nematocysts; (C) general view of nematocysts of secondary tentacles; (D) isorhizas of secondary tentacles; (E) eurytele of secondary tentacles; (F) discharged eurytele of secondary tentacles; (G) general view of nematocysts of the white spots; (H) birhopaloid of white spots; (I) discharged birhopaloid of white spots, with two dilatations (black arrows) close together; (J and K) nematocysts in the gastric filaments; (L and M) nematocysts on exumbrellar surface.

bifurcating at the stalk-calyx junction (Figs. 1, 2 and 5F), extending halfway up the calyx on the exumbrella along the interradial edge of each gonad, and sometimes also with lighter stripes running along both upper edges of the gonads near the calyx margin (Figs. 2B and 2C). Muscles whitish to colorless in the stalk and calyx (Figs. 5D and 5E).

**Cnidome.** Secondary tentacles with two types of nematocysts: isorhiza (abundant), length 9.54–13.50 μm (mean 12.11 μm, number of capsules measured $n = 62$), diameter 2.26–4.18 μm (mean 3.51 μm, $n = 62$); and microbasic eurytele (scarce), length 15.69–19.85 μm (mean 17.73 μm, $n = 24$), diameter 7.39–9.36 μm (mean 8.52 μm, $n = 24$) (Figs. 9A–9F).

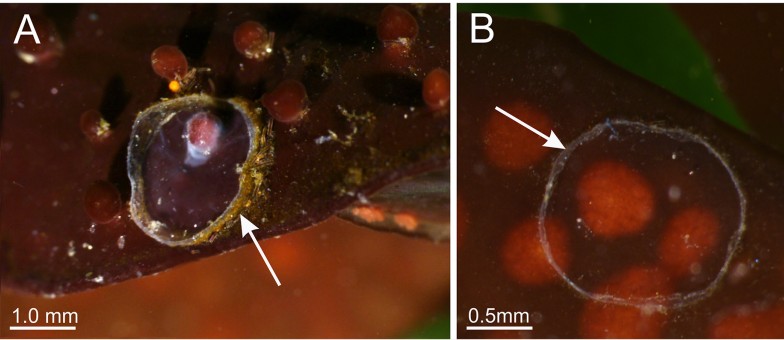

**Figure 10 Periderm of *Haliclystus sanjuanensis* n. sp.** Periderm (A and B), in the form of a thin transparent cup (white arrows) that encases the pedal disc, that has been left behind after the staur-omedusa *Haliclystus sanjuanensis* n. sp. moved.

White nematocyst spots with one type of nematocysts: birhopaloid type II (two dilatations close together) (abundant), length 11.63–17.13 μm (mean 14.60 μm, *n* = 50), diameter 7.95–11.39 μm (mean 9.96 μm, *n* = 50) (Figs. 9G–9I). Gastric filaments with one type of nematocysts: rhopaloid (not enough information to identify them as eurytele, stenotele, or birhopaloid) (abundant), length 7.10–10.70 μm (mean 8.90 μm, *n* = 11), diameter 4.13–6.80 μm (mean 5.47 μm, *n* = 11) (Figs. 9J and 9K). Exumbrella with one type of nematocysts: rhopaloid (not enough information to identify them as eurytele, stenotele, or birhopaloid), mean length 11.04 μm (*n* = 13), mean diameter 4.46 μm (*n* = 13) (Figs. 9L and 9M).

**Behavior.** This animal is almost always attached to the substrate by its pedal disc; some use of the anchors and secondary tentacles for anchorage has also been observed (Figs. 1 and 2). Only once or twice in many visits to the seashore have we seen a specimen free in the water. This stauromedusa produces a periderm in the form of a thin transparent cup that encases the pedal disc and is left behind when the staurozoan moves (Fig. 10); a new periderm is formed when the staurozoan reattaches. Small (usually arthropod) prey was captured by flexing (not contracting) one or more secondary tentacles on one or more adjacent arms, which immediately then bent in to the manubrium to transfer the captured prey to the mouth. Bending of an arm or arms was often accompanied by bending or twisting of the stalk. Spontaneous bending of single and multiple arms was typically observed. The anchors did not participate in feeding behaviors; their sole purpose appears to be as adhesive organs. One animal spawned spontaneously in the laboratory in May 2015, releasing many 35 μm diameter eggs in sticky mucus strands through its mouth opening of the manubrium (Fig. 11).

**Etymology.** The specific name, originally assigned by *Gellermann (1926)* in her M.S. Thesis, refers to the San Juan Archipelago, San Juan County, Washington State, USA, where this animal is abundant in some more-exposed areas, and where she studied it in detail. This location in NW Washington is approximately the epicenter of its presently-known distribution along NE Pacific shorelines and the location of some of the most-dense populations.

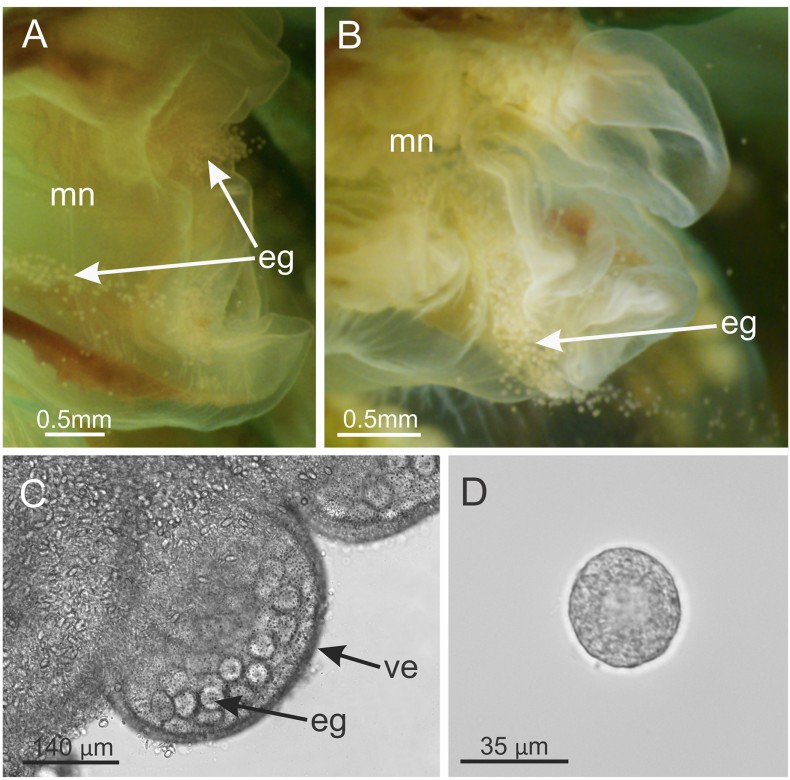

**Figure 11 *Haliclystus sanjuanensis* n. sp. spawning.** (A and B) Eggs release through month opening; (C) gonadal vesicles, with many eggs; (D) isolated egg. eg, eggs; ve, gonadal vesicles; mn, manubrium.

**Habitat.** On living eelgrass and surfgrass, *Zostera marina* and *Phyllospadix scouleri*, or on various species of red, green, and brown algae in the low intertidal to shallow subtidal, to at least 10 m deep in areas of moderate wave activity.

**Distribution.** Along the Pacific coast of North America at least from the eastern Aleutian Island to just north of Point Conception in central California, USA (Fig. 12; Table S1). It does not seem to occur south of Point Conception, as this species is not listed in the Southern California Association of Invertebrate Taxonomists species list for the Southern California Bight (*Cadien et al., 2018*), nor has it been seen by naturalist diver-photographers Kevin Lee, Merry Passage and Phil Garner in about 4,500 dives near Los Angeles (K. Lee & M. Passage, 2019, personal communications to C.E.M.).

On San Juan Island, Washington, USA, the type locality (Fig. 12), very small *H. sanjuanensis* n. sp. stauromedusae can be found beginning in approximately mid-April, and the last specimens persist into late November. No recognizable life history stages have been found from late fall until mid-spring.

**Genetic sequence data.** GenBank accession numbers: LSU (28S): KU308593, AY920782; SSU (18S): AF358102, KU308562, MN329739; SSU-ITS1-5.8S-ITS2-LSU: FJ874776, HM022143, HM022144, HM022145, MN329741; 16S: AY845339, HM022149, HM022150, HM022151, MN329738; COI: KU257477, MH242772, MN329740 (*Collins, 2002*; *Collins & Daly, 2005*; *Collins et al., 2006b*; *Miranda, Collins & Marques, 2010*;
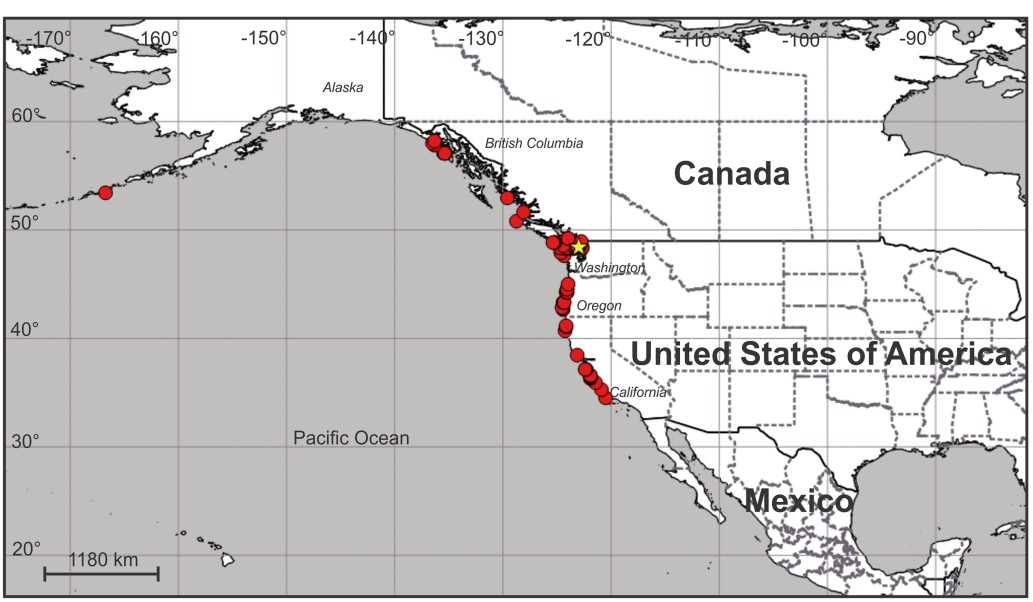

**Figure 12** Map of the geographic distribution of *Haliclystus sanjuanensis* n. sp. Star represents type locality. Map dataset made with Natural Earth. Free vector and raster map data from naturalearthdata. com.

*Miranda et al., 2016a*; *Westlake & Page, 2017*); Transcribed RNA sequences: HAHB01000001–HAHB01134305 (*Kayal et al., 2018*); Partial genome: JN700944 (*Kayal et al., 2012*).

**Remarks.** The genus *Haliclystus* is comprised of species with four interradial longitudinal muscles in the stalk (=peduncle) (Figs. 5, 6 and 7), a shared character of the suborder Myostaurida (see *Miranda et al., 2016a*). Within this suborder, *Haliclystus* is placed in the family Haliclystidae, characterized by the presence of perradial and interradial anchors (=rhopalioids, adhesive and sensorial structures derived from primary tentacles, located between adjacent pair of arms) (Figs. 1–3) (*Miranda et al., 2016a*; *Miranda & Collins, 2019*). The family Haliclystidae is also represented by species of the genera *Depastromorpha* Carlgren, 1935, *Depastrum* Gosse, 1858, *Halimocyathus James-Clark, 1863*, and *Manania James-Clark, 1863*. However, *Haliclystus* differs from these genera in lacking a claustrum, a tissue composed of a thin layer of mesoglea surrounded by a double layer of gastrodermis (*Miranda et al., 2016a*, *2016b*, *2017*). In addition, species of *Haliclystus* generally have four perradial chambers in the stalk (although this character is variable during development) (Figs. 5 and 6) (*Miranda et al., 2016a*, *2016b*) and the coronal muscle at the subumbrellar margin can be divided or entire (*Miranda et al., 2016a*). A monophyletic *Haliclystus* has been recovered in all phylogenetic analyses for the class (*Collins & Daly, 2005*; *Miranda, Collins & Marques, 2010*; *Miranda et al., 2016a*; *Holst & Laakmann, 2019*; *Holst, Heins & Laakmann, 2019*). However, in spite of recent efforts (*e.g.*, *Hirano, 1997*; *Miranda, Morandini & Marques, 2009*; *Miranda et al., 2016a*; *Kahn et al., 2010*), the taxonomy of the genus is still a challenge, as characters generally mentioned in the description of the species, such as number of tentacles and number of

chambers in the stalk are known to show intraspecific variation (*Uchida, 1929*; *Hirano, 1986*; *Miranda, Morandini & Marques, 2009*; *Kahn et al., 2010*).

*Haliclystus sanjuanensis* n. sp., discussed as a *nomen nudum* in different articles (see above), is herein properly described. *Haliclystus sanjuanensis* n. sp. is unique among other species of *Haliclystus* because of the arrangement of the bright-white nematocyst spots in perradii only, forming a conspicuous line along the perradial edges of gonads and rarely also a few near the perradial calyx margin (Figs. 1 and 4; see also Table S2; *Hirano, 1997*). *Haliclystus antarcticus* and *H. auricula*, the species phylogenetically most closely related to *H. sanjuanensis* n. sp. (*Miranda, Collins & Marques, 2010*; *Miranda et al., 2016a*; *Holst, Heins & Laakmann, 2019*), do not have these white nematocyst spots (see *Hirano, 1997*; *Miranda, Morandini & Marques, 2009*; *Miranda, Collins & Marques, 2013*). *Haliclystus borealis* Uchida, 1933, *Haliclystus californiensis Kahn et al., 2010*, *Haliclystus inabai* (Kishinouye, 1893), *Haliclystus monstrosus* (*Naumov, 1961*), *Haliclystus sinensis* Ling, 1937, *Haliclystus tenuis* Kishinouye, 1910, *H. salpinx*, and *H. octoradiatus* have white nematocyst spots in both perradii and interradii (see also additional differences in Table S2). There is no information concerning white spots for *Haliclystus kerguelensis* Vanhöffen, 1908. However, *H. sanjuanensis* n. sp. can be distinguished from *H. kerguelensis* based on the proportion between calyx and stalk and on the dark stripes (Table S2). Nevertheless, it is important to highlight that *H. kerguelensis* was described in 1908, based on a drawing and not specimens; *Kramp (1957)* reports three additional specimens collected in 1929–1930, but he does not include figures, so the validity of this species has never been tested.

The only *Haliclystus* species left for comparison is *H. stejnegeri*, which is morphologically very similar to *H. sanjuanensis* n. sp. (Table S2). However, although both species have white nematocyst spots found in perradii only, in *H. sanjuanensis* n. sp. the white spots are relatively abundant along the perradial edges of the gonads, running from near the margin to considerably deep into the subumbrella (Figs. 1 and 4), whereas in *H. stejnegeri* the white spots are relatively sparse and mainly found along the edge of gonads near the calyx margin (Fig. 13; Table S2). One additional difference is the pattern of pigmentation in the calyx (exumbrella). In *H. sanjuanensis* n. sp., eight dark stripes may be present, running halfway up the calyx along the interradial edges of gonads, which join as four stripes in the perradii running the length of the stalk; dark stripes may also be present near the calyx margin running along both edges of gonads (Fig. 2; Table S2). On the other hand, *H. stejnegeri* is characterized by its exclusive pattern with eight adradial pairs of dark stripes running the length of the calyx, and each two pairs united at the perradii of calyx base continuing as four dark stripes running down the stalk; the calyx margin is also tinted in the same color (*Kishinouye, 1899*; Fig. 13; Table S2). Unfortunately, preserved stauromedusae frequently lose their live coloration, so this character can be difficult to assess in preserved specimens.

Although some morphological characters used to differentiate species of *Haliclystus* vary with development (*Uchida, 1929*; *Hirano, 1986*; *Miranda et al., 2016b*), the current valid species have good support based on molecular evidence. Nine out of 13 described species of *Haliclystus* have been phylogenetically placed (*Miranda et al., 2016a*; *Holst &*

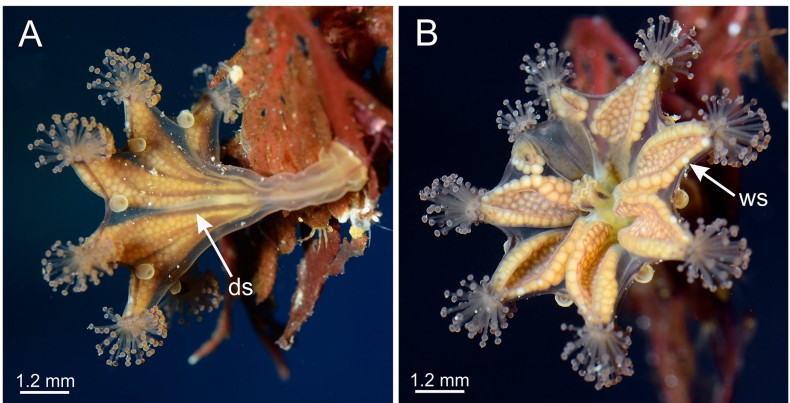

**Figure 13 General morphology of *Haliclystus stejnegeri*.** (A) Lateral view, with dark stripes pattern (see Table S2); (B) oral view, with few white nematocyst spots at perradii. ds, dark stripes; ws, white nematocyst spot. Images courtesy of Karen Sanamyan and Nadya Sanamyan (locality: Avacha Bay, Kamchatka, Russia).

*Laakmann, 2019*; *Holst, Heins & Laakmann, 2019*), including *H. sanjuanensis* n. sp.. Interestingly, although morphologically similar, *H. sanjuanensis* n. sp. and *H. stejnegeri* did not come out as closely related species (*Miranda et al., 2016a*; *Holst & Laakmann, 2019*; *Holst, Heins & Laakmann, 2019*). *Haliclystus sanjuanensis* n. sp. proved to be more closely related to *H. antarcticus* and *H. auricula* (*Miranda, Collins & Marques, 2010*; *Kayal et al., 2018*; *Holst, Heins & Laakmann, 2019*), despite the huge geographical gap between the known distributions of these species: Western United States and Canada for *H. sanjuanensis* n. sp. (Fig. 12); Southern Chile and Argentina and the Antarctic Peninsula for *H. antarcticus* (*Miranda et al., 2018*); and Eastern United States and Europe for *H. auricula* (*Miranda et al., 2018*).

The geographic distribution of *H. sanjuanensis* n. sp. in the North Pacific Ocean begins at about 34.5° N in central California and extends up at least to Southeast Alaska (about 58.2° N) and also west into the Aleutian Islands (to at least 166.8° W) (Fig. 12). It may or may not overlap *H. stejnegeri* at the upper end of its range in the NE Pacific, but then *H. stejnegeri* appears to swing across and down the NW Pacific to the Kurile Islands and to northern Japan (*Miranda et al., 2018*). *Haliclystus californiensis* is a very distinctive orangey-red species that is sympatric with *H. sanjuanensis* n. sp., occurring from southern California to at least central British Columbia (*Kahn et al., 2010*; our studies, see also Table S2). Four more described species of *Haliclystus*, easily separated from *H. sanjuanensis* n. sp. because of the pattern of white nematocyst spots and the number of secondary tentacles on each arm in adult stauromedusae (see Table S2), are found in NW Pacific waters of China, Far East Russia, Japan, and Korea (*H. sinensis, H. borealis, H. tenuis*, and *H. inabai*) (see *Miranda et al., 2018*).

At least one more possibly undescribed species of *Haliclystus* (*Gellermann, 1926*) occurs at about the midpoint of the south to north distribution of *H. sanjuanensis* n. sp., from the San Juan Islands and southern Vancouver Island to at least SE Alaska and possibly all the way to the Kuriles, northern Japan, and mainland far-east Russia, where a look-alike has been described as *H. monstrosus* (*Naumov, 1961*; *Sheiko & Stepanyants, 1990*). This

possibly undescribed species of *Haliclystus* is also similar to *H. salpinx*, a species found in the North Atlantic (*James-Clark, 1863*; *Berrill, 1962*; but see also *Sheiko & Stepanyants, 1990*), raising the hypothesis that all records of *H. salpinx* in Washington State (*Otto, 1976*, *1978*; *Mills, 1987*; *Strathmann, 1987*; *Wrobel & Mills, 2003*; *Mills & Larson, 2007*) are misidentifications, but we need to complete further work to be sure. We are in the process of collecting more material for DNA and morphological comparisons to determine how many other *Haliclystus* species are present in the boreal North Pacific.

## ACKNOWLEDGEMENTS

We thank the scientists, our colleagues, and the staff at the Friday Harbor Labs for their intellectual support and friendship over the years and for hosting our international collaborative study of *Haliclystus sanjuanensis* n. sp. in May 2015. James T. (Jim) Carlton generously read through many early editions of books about the natural history of the California seashore in the early twentieth century and sent us scans of all pages mentioning stauromedusae. C.E.M. had many enjoyable discussions with West Coast photographers including Ron Larson, Neil McDaniel, Andy Lamb, Ron Shimek, Jeff Goddard, Gary McDonald, Steve Lonhart, Allison Gong, Jeff Mondragon, Gustav Paulay, Nancy Treneman, Dave Behrens, and several photographers who posted images on iNaturalist.com, identifying their photographs of *Haliclystus* as she developed a detailed picture of the range of this species. We thank Gustav Paulay, Ron Larson, and Neil McDaniel for allowing us to use their photographs of living *Haliclystus sanjuanensis* n. sp. for this description (Figs. 1, 2); Gustav Paulay's photographs in Fig. 2 were taken during the summer 2017 Bioblitz organized and funded by the Hakai Institute, at Calvert Island, BC. We also thank Karen Sanamyan and Nadya Sanamyan for kindly allowing us to use their photographs of living *Haliclystus stejnegeri* in Fig. 13. Henry Choong, Curator of Invertebrate Zoology at the Royal British Columbia Museum in Victoria, sent us photographs of RBCM specimen #015-00395 collected in May 2015 in Haida Gwaii, and subsequently hosted C.E.M. for an in-person visit of the RBCM collection of *Haliclystus* stauromedusae. We are very grateful to the editor Timothy Collins, to the reviewer Dr. André Carrara Morandini, and to two anonymous reviewers who helped to improve the quality of this manuscript.

### Funding

This study was supported by PRPq/UFMG ADRC (grant number 26048 * 132) (Lucília S. Miranda) and by FAPESP (grant number 2015/23695-0) (Lucília S Miranda). Hannah Westlake's participation in this project was funded by a Canadian NSERC Discovery Grant to Louise R. Page. The funders had no role in study design, data collection and analysis, decision to publish, or preparation of the manuscript.

## Grant Disclosures

The following grant information was disclosed by the authors:
PRPq/UFMG ADRC: 26048 * 132.
FAPESP: 2015/23695-0.
Canadian NSERC Discovery Grant to Louise R. Page.

## Competing Interests

The authors declare that they have no competing interests.

## Author Contributions

- Claudia E. Mills conceived and designed the experiments, performed the experiments, analyzed the data, prepared figures and/or tables, authored or reviewed drafts of the article, and approved the final draft.
- Hannah Westlake conceived and designed the experiments, performed the experiments, analyzed the data, prepared figures and/or tables, authored or reviewed drafts of the article, and approved the final draft.
- Yayoi M. Hirano conceived and designed the experiments, performed the experiments, analyzed the data, prepared figures and/or tables, authored or reviewed drafts of the article, and approved the final draft.
- Lucília S. Miranda conceived and designed the experiments, performed the experiments, analyzed the data, prepared figures and/or tables, authored or reviewed drafts of the article, and approved the final draft.

## Field Study Permissions

The following information was supplied relating to field study approvals (*i.e.*, approving body and any reference numbers):

The waters of San Juan County and Cypress Island (the San Juan Archipelago), Washington, are a designated Marine Biological Preserve, with the University of Washington Friday Harbor Laboratories as the managing agency (Washington State House Bill 68, 1923; R.C.W.28.77.230,1969; Revision R.C.W.28B.20.320, 2023); since 1923, the Director of the Friday Harbor Laboratories grants approval of scientific collection of marine plants and animals within the Marine Biological Preserve and C.E.M. has been granted such permission for collection of the marine animals reported in this study. Staurozoa were collected in Victoria, British Columbia, Canada, under Scientific License XR 30 2013-15 issued by the Department of Fisheries and Oceans Canada to Professor Louise Page, academic advisor of H.W. during those years.

## Data Availability

The raw data is available in the Supplemental Files.

## New Species Registration

The following information was supplied regarding the registration of a newly described species:

Publication LSID: urn:lsid:zoobank.org:pub:BF343A8D-DB31-4150-840C-6CDA5A5B121E.

Genus name: https://zoobank.org/NomenclaturalActs/EA198E96-20F7-4810-A972-3EE68DC73CFA.

Species name: urn:lsid:zoobank.org:act:76EEDCEB-02E9-497B-80E9-3473B506225D.

## Supplemental Information

Supplemental information for this article can be found online at http://dx.doi.org/10.7717/peerj.15944#supplemental-information.

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
