# Peer review of "Description of a common stauromedusa on the Pacific Coast of the United States and Canada, Haliclystus sanjuanensis new species (Cnidaria: Staurozoa)"

_PeerJ, doi:10.7717/peerj.15944_

## Round 0.1 · original submission · Minor Revisions

In this manuscript the authors provide a formal description of a commonly found stalked jellyfish found on the Pacific coast of the US and Canada. The three external reviews are all favorable, and consider this a valuable taxonomic contribution for the staurozoa. I agree with their assessment, and suggest that it be accepted pending minor revisions. Please address each of the reviewer's comments in your cover letter accompanying the revised manuscript. I have sent a copy of the manuscript in Word with my suggested revisions to facilitate the process

Reviewer 1 ·

Basic reporting

Incredibly thorough species description. The synonomy list is fantastic and all the relevant literature is cited.

Experimental design

Straight forward description of a species that has been known. Extremely well documented, and rigorous. The description covers gross anatomy, histology, cnidome, genetics, behavior, distribution.

Validity of the findings

There is no doubt about this being a new species. It is very well explained and documented. Data are all made available.

Additional comments

An excellent piece of work. This is a rare review in that I think the paper can be accepted as is. One small question for the authors to consider.

They describe the group as "rare", which may indeed be true. They are also well camouflaged and not particularly large. It may be that they are rare and/or often overlooked. It is a small group among perhaps a million eukaryotic marine species, so perhaps that is the perspective the authors had in mind. As they note, however, from time to time populations of staurozoan species are locally abundant.

·

Basic reporting

The manuscript is relevant in the sense that it provides stability to the Zoological Nomenclature by offering a proper description of a commonly found and pictured staurozoan species. In general, the text is well organized and concise with clear objectives; language and grammar used is very good. The authors are acknowledged in providing the extensive synonymy list bringing light into a cloudy and, sometimes, confusing identification of this new species (considering that there are other similar and co-occurring species).

Experimental design

The methods presented are in accordance with the recent and updated literature of the group. The citations are broad and cover all the knowledge of the group, including ancient and brand new sources.

Validity of the findings

The results are adequate for systematics and taxonomy studies, and the findings are important for helping organize the classification of the group. Images, tables and supplementary material are all important for the understanding of the new species description and comparison with other species of the genus.
There are some minor comments and suggestions on the PDF manuscript file.

Additional comments

My decision is to accept the manuscript after minor corrections, with no need of additional evaluation.
If the journal policy permits, I have no restrictions that my name is informed to the authors.

Reviewer 3 ·

Basic reporting

Mills et al provide a formal description of Haliclystus sanjuanensis, a staurozoan cnidarian. H. sanjuanensis has been a nomen nudum, potentially creating issue with the misapplication of the name in the literature. The formal description of H. sanjuanensis provided here is, in my opinion, important to ensure reliable and stable identification of the species and its distribution.

Experimental design

The authors collected numerous specimens over the years which they studied and deposited a large number of specimens as type material to several museums/collections. I am not an expert on staurozoan systematics, but the morphological characterization of H. sanjuanesis seems extensive and comprehensive. H. sanjuanensis is also clearly delineated from its congeners in the current description.

Validity of the findings

The authors conducted a comprehensive investigation of collected specimens, photographs and the literature to substantiate their findings and provide a robust description of H. sanjuanensis. The synonymy list is extensive and should be quite useful to others in the field.

Additional comments

l. 45: Consider deleting "the reader will discover...".

l. 50: Instead of "this animal" use the species name; be specific.

ll. 75 - 92: I think several of these paragraphs may be merged; each paragraph is rather short and the concepts are connected.

l. 94: Provide the article of the code instead of the URL. URLs may change, making it difficult for the reader to find the correct section of the code in the future.

l. 101: Can you be more specific? You state that you looked at hundreds of specimens. A ball-park number would be useful.

ll. 138 - 151: I think these paragraphs could be merged into one larger one.

l. 160: "The slides are deposited..." should be were deposited.

l. 347: Should the heading contain all caps? I suggest changing to normal font.

l. 521: "Gene sequences"; technically not all the sequence data referenced encode genes (eg, rRNA is not translated). Consider changing to genetic sequence data instead.

ll. 552 - 555: Consider abbreviating genus names consistently. As is, there is a mix of abbreviations and fully written names of the same genus in this section.

l. 577: "tricky"; can you be more specific what it means for a character to be tricky? The characters are homoplasious, difficult to differentiate...

l. 582: "molecularly very similar"; do you mean closely related or small genetic distance in some marker (which marker)? Be specific.

l. 599: "picks up"; this seems rather colloquial. Maybe "starts occurring around" or a similar phrasing would be better.

---

## Round 0.2 · accepted · Accept

The authors have done a nice job of responding to the comments of the external reviewers, who recommended acceptance or minor revisions. In my opinion, the manuscript is now acceptable for publication.